# Effects of footshock stress on social behavior and neuronal activation in the medial prefrontal cortex and amygdala of male and female mice

**Mariia Dorofeikova**[1,2☯], **Chandrashekhar D. Borkar**[1,2☯], **Katherine Weissmuller**[3], **Lydia Smith-Osborne**[1,4], **Samhita Basavanhalli**[3], **Erin Bean**[3], **Avery Smith**[3], **Anh Duong**[1,3], **Alexis Resendez**[1,2], **Jonathan P. Fadok**[1,2]*

1 Department of Psychology, Tulane University, New Orleans, LA, United States of America, 2 Tulane Brain Institute, Tulane University, New Orleans, LA, United States of America, 3 Neuroscience Program, Tulane University, New Orleans, LA, United States of America, 4 Tulane National Primate Research Center, Covington, LA, United States of America

☯ These authors contributed equally to this work.
* jfadok@tulane.edu

**Data Availability Statement:** Here is the DOI for the data in our manuscript: https://doi.org/10.7910/DVN/1RAQON.

## Abstract

Social behavior is complex and fundamental, and its deficits are common pathological features for several psychiatric disorders including anxiety, depression, and posttraumatic stress disorder. Acute stress may have a negative impact on social behavior, and these effects can vary based on sex. The aim of this study was to explore the effect of acute footshock stress, using analogous parameters to those commonly used in fear conditioning assays, on the sociability of male and female C57BL/6J mice in a standard social approach test. Animals were divided into two main groups of footshock stress (22 male, 24 female) and context exposed control (23 male and 22 female). Each group had mice that were treated intraperitoneally with either the benzodiazepine—alprazolam (control: 10 male, 10 female; stress: 11 male, 11 female), or vehicle (control: 13 male, 12 female; stress: 11 male, 13 female). In all groups, neuronal activation during social approach was assessed using immunohistochemistry against the immediate early gene product cFos. Although footshock stress did not significantly alter sociability or latency to approach a social stimulus, it did increase defensive tail-rattling behavior specifically in males ($p = 0.0022$). This stress-induced increase in tail-rattling was alleviated by alprazolam ($p = 0.03$), yet alprazolam had no effect on female tail-rattling behavior in the stress group. Alprazolam lowered cFos expression in the medial prefrontal cortex ($p = 0.001$ infralimbic area, $p = 0.02$ prelimbic area), and social approach induced sex-dependent differences in cFos activation in the ventromedial intercalated cell clusters ($p = 0.04$). Social approach following stress-induced cFos expression was positively correlated with latency to approach and negatively correlated with sociability in the prelimbic area and multiple amygdala subregions (all $p < 0.05$). Collectively, our results suggest that acute footshock stress induces sex-dependent alterations in defensiveness and differential patterns of cFos activation during social approach.

**Funding:** This work was supported by Tulane University, the Louisiana Board of Regents through the Board of Regents Support Fund (LEQSF(2018-21)-RD-A-17), and the National Institute of Mental Health of the National Institutes of Health under award number R01MH122561 to JPF. The content is solely the responsibility of the authors and does not necessarily represent the official views of the National Institutes of Health. The funders had no role in study design, data collection and analysis, decision to publish, or preparation of the manuscript. There was no additional external funding received for this study.

**Competing interests:** The authors have declared that no competing interests exist.

## Introduction

Social behavior is important for survival, and social deficits are common pathological features for a variety of mental illnesses including social anxiety disorder, depression, and post-traumatic stress disorder [1, 2]. More women than men suffer from these disorders, yet there is a paucity of data on sex differences in social behavior after stress [3]. There are data suggesting that childhood trauma leads to more aggressive behavior in men and more social withdrawal and avoidance in women [4], and women with social anxiety disorder report greater clinical severity, which may be associated with stressful life experiences [5]. Therefore, understanding sex differences in the effects of traumatic stress on social behavior, as well as the underlying neural substrates that potentially control this behavior, has important translational relevance.

Animal models of stress and trauma show alterations in many aspects of behavior [6]. In general, both acute and chronic stress have been found to lead to social fear and withdrawal in rodents [1, 7]. One common model of traumatic stress in rodents is footshock exposure [8]. However, there is a lack of data investigating whether footshock stress influences mouse social behavior in a sex-dependent manner. In rats, footshock stress has been shown to induce social avoidance in animals with elevated levels of fear generalization [9], and impair the behavioral response to a social-paired compartment [10]. Moreover, intense footshock (2 mA, 10 s), followed by situational reminders, elicits impairments in social interaction in female rats [11].

It is also unclear whether there are sex differences in the activation of brain regions responsive to social encounters following acute stress. Several lines of evidence suggest that neural activity in the medial prefrontal cortex (mPFC) is important for social behavior and subregions of the mPFC are differentially activated following social interaction. Neurons in the infralimbic cortex (IL) are preferentially activated in response to social cues compared to neurons in the prelimbic cortex (PL) [12], and a social neural ensemble within the IL may contribute to the social buffering of fear after fear conditioning [2, 13].

Several subnuclei of the amygdala have also been implicated in social behavior. The role of the basolateral amygdala in social interaction following different stress paradigms has been well-established [14–16]. For example, activation of the basolateral amygdala leads to reduced social interaction in a social interaction test [3, 17]. The medial nucleus of the amygdala (MeA) is also involved in both social behaviors and responses to stressors [18]. cFos expression in the intercalated nucleus of the amygdala (ITC) is increased during social buffering in rats [19], and social interaction is impaired in mice with altered migration and differentiation of ITCs [20]. Among the amygdala regions implicated in social behavior, the central nucleus of the amygdala (CeA) is relatively unexplored. There is recent evidence, however, that CeA circuits may be linked to sociability, and some manipulations of CeA activity impact social behavior [21–23].

Thus, multiple brain regions are involved in social cognition [12, 14], but the differences in social approach or avoidance behavior following acute footshock stress are still poorly understood. In the current study, our goal was to assess the extent to which there are sex differences in sociability using a social approach test that eliminates the possibility of direct physical aggression. We hypothesized that two days of footshock stress would negatively affect sociability, and that those changes might depend on sex and be reversible with the fast-acting benzodiazepine alprazolam, which is used in short-term management of anxiety disorders. Additionally, we aimed to determine patterns of neuronal activation in the mPFC, CeA, MeA, and ventral ITC associated with social behaviors using expression of the immediate early gene cFos.

## Materials and methods

### Animals

2–4-month-old male and female C57BL/6J mice were obtained from the Jackson Laboratory (Bar Harbor, ME, Stock No: 000664) and housed on a 12 h light/dark cycle with *ad libitum* access to water and chow under standard laboratory conditions. Mice were individually housed for 7 days before the start of and all throughout the experiments, since single housing avoids intermale aggression and social dominance-induced behavioral changes [24]. Experiments were performed during the light phase. All animal procedures were performed in accordance with institutional guidelines and were approved by the Institutional Animal Care & Use Committee of Tulane University (ethics approval protocol ID– 1013). Unfamiliar strain-, sex- and age-matched mice (N = 33) were used as stimulus mice during social approach tests.

### Groups

A total of 45 males and 46 females were separated into the following groups:

1) Control males treated with vehicle, N = 13; 2) Control females treated with vehicle, N = 12; 3) Control males treated with alprazolam, N = 10; 4) Control females treated with alprazolam, N = 10; 5) Stressed males treated with vehicle, N = 11; 6) Stressed females treated with vehicle, N = 13; 7) Stressed males treated with alprazolam, N = 11; 8) Stressed females treated with alprazolam, N = 11.

### Footshock stress exposure

Footshock exposure or control context exposure was conducted in standard mouse operant conditioning chambers (ENV-307W, Med Associates, Inc., St. Albans, VT) enclosed within sound- and light-attenuating cubicles (ENV-022MD, Med Associates, Inc., St. Albans, VT). The chambers were connected to a computer through an interface and controlled by MED-PC software. The chamber was equipped with a grid floor and a house light, which was cleaned using 70% ethanol.

Seven days after single housing, mice underwent footshock exposure for two consecutive days. Each of the shock sessions included five 1 s, 0.9 mA footshocks presented with a 120 s average pseudorandom intertrial interval (range 90–150 s), totaling 800 s in the chamber. The intensity of footshock was chosen based on previous studies [25, 26]. Mice in the control group were exposed to the same chambers for the same period but did not experience footshock.

### Social approach test

The day after footshock exposure, the mice underwent the social approach test in a square 46 x 46 X 38 cm arena constructed from sheets of white plexiglass. Behavioral videos were recorded using a digital camera (Allied Vision "Pike" camera, Germany) and Plexon Studio tracking software (Plexon, Dallas, TX). Tests were conducted under dim (10.6 lux) white fluorescent lighting. Stimulus mice were single housed for 3 days before tests. Each of the stimulus mice interacted with three experimental mice with at least 30 min between tests. Experimental mice were perfused 90 min after the test to assess cFos expression.

An indirect social interaction method was chosen to avoid physical aggression between male mice. For the first 3 min, mice were allowed to explore the open arena with two rectangular (15 X 5 X 6 cm) or circular (8 cm diameter, 10 cm high) metallic mesh boxes located in opposing corners 5 cm away from the walls. After the initial exploration, an unfamiliar, untreated stimulus mouse was put underneath one of the boxes. Behavior was recorded for an

additional 5 minutes, and sociability was scored using time spent sniffing the mesh box containing the stimulus mouse as a percentage of total box interaction time (mouse preference, %), the latency to approach the stimulus mouse, and the number of defensive tail rattles. Total exploration of the mesh boxes was also scored in seconds to assess general activity. All behavioral measurements were scored by an observer blinded to condition. Consistent with other social approach scoring protocols [27], sniffing directed to the upper and top part of the mesh boxes, sniffing of feces, bar biting and circulating around the corral without sniffing, were not scored as social approach.

## Alprazolam treatment

Alprazolam (Sigma-Aldrich, St. Louis, MO) was dissolved in a drop of Tween 80 (Merck, Germany) and saline was added to make a final dose of 0.25 mg/kg. This dose was shown to have anxiolytic effects [28] with minimal motor impairment in C57BL/6J mice [29]. Tween 80 + saline solution was used for vehicle injections. Solutions were administered at 10 ml/kg volume, intraperitoneally, 30 min before social approach tests.

## Histology

Following testing, mice were anesthetized with 2,2,2-tribromoethanol (240 mg/kg, ip, Sigma) and subsequently transcardially perfused with 4% paraformaldehyde in phosphate-buffered saline (PBS). Subjects for cFos analysis were chosen randomly from the respective behavioral cohorts and balanced across the groups. cFos expression was assessed in mice that were perfused 90 min after the social approach test. Fixed brains were cut on a compresstome vibrating microtome (Precisionary, Greenville, NC) in 80 μm coronal slices.

Antibody staining was performed on free-floating tissue sections. After 3 x 10 min washes with 0.5% PBST slices were put in 5% donkey serum for 2 hours. Sections were then incubated overnight in primary rabbit anti-cFos antibody (dilution 1:1500; #226 003, Synaptic Systems, Germany) at 4°C. On the next day sections were washed in 0.5% PBST (3 X 10 min), and then went through a 2 hr incubation with secondary donkey anti-rabbit antibody AlexaFluor 488 (dilution 1:500; #A-21206, Thermo Fisher Scientific, Waltham, MA) at 4°C. After 3 x 10 min washes in PBS slices were mounted with mounting medium with DAPI (Biotium, Fremont, CA).

Images were obtained using an AxioScan.Z1 slide-scanning microscope (Zeiss, Germany) and a Nikon A1 Confocal microscope (Nikon, Japan). cFos-positive nuclei were quantified using Fiji ImageJ software (NIH, Bethesda, USA), and averaged for each animal. A blinded observer quantified cFos expression in 2–5 slices per structure per mouse.

## Statistical analysis

Data were analyzed using Prism 9 (GraphPad Software, San Diego, CA). The definition of statistical significance was $p \leq 0.05$. Two male mice that showed no interaction with either mesh box during the social approach test were excluded from analysis. Because the data were non-continuous, tail-rattling was analyzed using Fisher's exact test. To assess the interaction of factors, a 3-way ANOVA was used. If a significant effect was detected, Sidak's multiple comparisons test was used, because it assumes independent comparisons and has more power than the Bonferroni method. Correlations between cFos expression and behavior were analyzed for control (N = 16) and stressed (N = 16) mice using pooled data from alprazolam and vehicle treatment groups of both sexes. The mice having cFos data for all brain regions were randomly selected for correlations with their behavior. Using the Kolmogorov-Smirnov test, the data were determined to be non-parametric, so Spearman's correlation coefficient was

used for this analysis. Because there was a Drug x Sex interaction and a main effect of Drug in cFos expression in the IL, we did not include these data in the correlation analysis. For the sake of clarity, we report the results of the interaction tests, the significant simple main effects, and the significant post-hoc tests in the main text. The results of all tests are reported in **Table 1**. All statistical tests were two-tailed.

## Results

### Effects of footshock stress and alprazolam treatment on social behavior

Following two days of footshock stress or context exposure, male and female mice were allotted to the vehicle and alprazolam treatment groups. These mice were then subjected to a social approach test designed to measure sociability (**Fig 1A and 1B**). A three-way ANOVA revealed that there was no statistically significant interaction between the effects of either stress, alprazolam, sex or their interactions on sociability (**Fig 1C**; Sex x Stress x Drug, $F_{(1,83)} = 0.059$; $p = 0.80$) or latency to approach the social stimulus (**Fig 1D**; Sex x Stress x Drug, $F_{(1,83)} = 0.24$; $p = 0.62$).

Further, we applied Fisher's exact test to analyze noncontinuous tail-rattling data. Interestingly, significantly higher number of stressed males displayed tail-rattling behavior than control males during the social approach test ($p = 0.002$), while females in both groups displayed equivalent levels of tail rattling (**Fig 1E**; $p > 0.99$). On the other hand, alprazolam treatment significantly reduced the number of males showing tail-rattling behaviors ($p = 0.03$) but did not affect tail-rattling in females. In the control vehicle condition, females show higher tail-rattling than males, although this difference did not reach statistical significance ($p = 0.14$). Exploratory behavior, measured as total interaction time with both mesh boxes, was unaffected by footshock stress or alprazolam treatment (**Fig 1F**; 3-way ANOVA, sex X stress X drug, $F_{(1, 83)} = 0.03$, $p = 0.86$).

### cFos expression analysis

We next quantified expression of cFos in several brain regions involved in the regulation of social behavior (**Fig 2A and 2B**). A three-way ANOVA was performed to test for the effects of sex, stress, and alprazolam treatment on cFos expression in the IL, PL, CeA, ventromedial ITC, and MeA (N = 13 control vehicle (5 male, 8 female), N = 10 control alprazolam (5 male, 5 female), N = 14 stress vehicle (5 male, 9 female), N = 12 stress alprazolam (6 male, 6 female). There was no significant three-way interaction between the effects of these variables on cFos expression in any of the brain areas analyzed (see **Table 1**). There was a significant sex by alprazolam interaction effect on cFos expression in IL (**Fig 2C**; $F_{(1,41)} = 5.78$, $p = 0.02$) and a main effects analysis showed that alprazolam treatment significantly reduced cFos expression in the IL (drug effect, $F_{(1,41)} = 12.6$, $p = 0.001$). Post hoc analysis showed that control females injected with alprazolam had significantly fewer cFos+ cells in the IL compared to vehicle-injected female controls (Sidak's multiple comparisons test, $p = 0.03$). There was a main effect of drug on cFos expression in the PL (**Fig 2D**; drug effect, $F_{(1,33)} = 6.55$, $p = 0.02$). There were no significant simple main effects of sex, stress, or drug on cFos expression in the capsular subdivision of the CeA (**Fig 2E**), the medial (**Fig 2F**, sex, $F_{(1, 34)} = 4.11$, $p = 0.0505$) or lateral subdivision of the CeA (**Fig 2G**, sex, $F_{(1, 34)} = 3.96$, $p = 0.0546$), or in the medial amygdala (**Fig 2I**). There was a significant effect of sex on cFos expression in the ventromedial intercalated nucleus of amygdala, with greater expression levels in males (**Fig 2H**; sex effect, $F_{(1,34)} = 4.53$, $p = 0.04$).

**Table 1. Results of statistical analyses.**

**Effects of alprazolam x footshock stress x sex: 3-way ANOVA**

| Source | Sum of squares | DFn | DFd | F | p |
|---|---|---|---|---|---|
| Fig 1A Sociability (Mouse preference, %) | | | | | |
| Alprazolam*Stress*Sex | 23.8 | 1 | 83 | 0.059 | 0.80 |
| Alprazolam*Stress | 757 | 1 | 83 | 1.88 | 0.17 |
| Alprazolam*Sex | 373 | 1 | 83 | 0.92 | 0.33 |
| Stress*Sex | 292 | 1 | 83 | 0.72 | 0.39 |
| Alprazolam | 5.22 | 1 | 83 | 0.013 | 0.90 |
| Stress | 83.2 | 1 | 83 | 0.20 | 0.65 |
| Sex | 75.4 | 1 | 83 | 0.18 | 0.66 |
| Fig 1B Latency to approach | | | | | |
| Alprazolam*Stress*Sex | 1765 | 1 | 83 | 0.24 | 0.62 |
| Alprazolam*Stress | 4503 | 1 | 83 | 0.61 | 0.43 |
| Alprazolam*Sex | 8301 | 1 | 83 | 1.14 | 0.28 |
| Stress*Sex | 11760 | 1 | 83 | 1.61 | 0.20 |
| Alprazolam | 1570 | 1 | 83 | 0.21 | 0.64 |
| Stress | 15040 | 1 | 83 | 2.06 | 0.15 |
| Sex | 7434 | 1 | 83 | 1.02 | 0.31 |
| Fig 1D Exploration | | | | | |
| Alprazolam*Stress*Sex | 5.7 | 1 | 83 | 0.03 | 0.86 |
| Alprazolam*Stress | 15 | 1 | 83 | 0.09 | 0.77 |
| Alprazolam*Sex | 104 | 1 | 83 | 0.60 | 0.44 |
| Stress*Sex | 23 | 1 | 83 | 0.13 | 0.71 |
| Alprazolam | 146 | 1 | 83 | 0.85 | 0.36 |
| Stress | 0.91 | 1 | 83 | 0.005 | 0.94 |
| Sex | 55 | 1 | 83 | 0.32 | 0.57 |
| Fig 1C Tail rattling | | | | | |
| Contingencies | Yes | No | | Fisher's exact test | |
| Male vehicle control vs Male vehicle stress | 1 vs 8 | 12 vs 3 | | p < 0.0022* | |
| Female vehicle control vs Female vehicle stress | 4 vs 5 | 7 vs 8 | | p > 0.9999 | |
| Male vehicle control vs Female vehicle control | 1 vs 4 | 12 vs 7 | | p < 0.1421 | |
| Male veh stress Male alp stress | 8 vs 2 | 3 vs 9 | | p < 0.0300* | |
| Female veh stress Female alp stress | 5 vs 4 | 8 vs 7 | | p > 0.9999 | |

**Effects of alprazolam x footshock stress x sex on cFos: 3-way ANOVA**

| Source | Sum of squares | DFn | DFd | F | p |
|---|---|---|---|---|---|
| Fig 2C IL PFC | | | | | |
| Alprazolam*Stress*Sex | 579981 | 1 | 41 | 2.43 | 0.13 |
| Alprazolam*Stress | 172328 | 1 | 41 | 0.72 | 0.40 |
| Alprazolam*Sex | 1378855 | 1 | 41 | 5.78 | **0.02*** |
| Stress*Sex | 169279 | 1 | 41 | 0.71 | 0.40 |
| Alprazolam | 3014936 | 1 | 41 | 12.6 | **0.001*** |
| Stress | 2231 | 1 | 41 | 0.01 | 0.92 |
| Sex | 538912 | 1 | 41 | 2.26 | 0.14 |

*(Continued)*

**Table 1.** (Continued)

| Fig 2D PL PFC | | | | | |
|---|---|---|---|---|---|
| Alprazolam*Stress*Sex | 657.3 | 1 | 33 | 0.008 | 0.93 |
| Alprazolam*Stress | 322417 | 1 | 33 | 3.86 | 0.06 |
| Alprazolam*Sex | 169139 | 1 | 33 | 2.03 | 0.16 |
| Stress*Sex | 7228 | 1 | 33 | 0.09 | 0.77 |
| Alprazolam | 546812 | 1 | 33 | 6.55 | **0.02**\* |
| Stress | 3566 | 1 | 33 | 0.04 | 0.84 |
| Sex | 29411 | 1 | 33 | 0.35 | 0.56 |
| Fig 2E Capsular CeA | | | | | |
| Alprazolam*Stress*Sex | 24040 | 1 | 34 | 0.55 | 0.46 |
| Alprazolam*Stress | 19319 | 1 | 34 | 0.45 | 0.51 |
| Alprazolam*Sex | 54202 | 1 | 34 | 1.25 | 0.27 |
| Stress*Sex | 45335 | 1 | 34 | 1.05 | 0.31 |
| Alprazolam | 38512 | 1 | 34 | 0.89 | 0.35 |
| Stress | 36099 | 1 | 34 | 0.83 | 0.37 |
| Sex | 82985 | 1 | 34 | 1.91 | 0.18 |
| Fig 2F Medial CeA | | | | | |
| Alprazolam*Stress*Sex | 5538 | 1 | 34 | 0.92 | 0.34 |
| Alprazolam*Stress | 3419 | 1 | 34 | 0.57 | 0.46 |
| Alprazolam*Sex | 8969 | 1 | 34 | 1.49 | 0.23 |
| Stress*Sex | 8175 | 1 | 34 | 1.35 | 0.25 |
| Alprazolam | 15566 | 1 | 34 | 2.59 | 0.12 |
| Stress | 307.6 | 1 | 34 | 0.05 | 0.83 |
| Sex | 24836 | 1 | 34 | 4.11 | 0.05 |
| Fig 2G Lateral CeA | | | | | |
| Alprazolam*Stress*Sex | 6744 | 1 | 34 | 0.38 | 0.54 |
| Alprazolam*Stress | 10219 | 1 | 34 | 0.58 | 0.45 |
| Alprazolam*Sex | 762.9 | 1 | 34 | 0.04 | 0.84 |
| Stress*Sex | 11.84 | 1 | 34 | 0.0005 | 0.98 |
| Alprazolam | 11499 | 1 | 34 | 0.65 | 0.42 |
| Stress | 3797 | 1 | 34 | 0.22 | 0.64 |
| Sex | 69608 | 1 | 34 | 3.97 | 0.05 |
| Fig 2H ITC | | | | | |
| Alprazolam*Stress*Sex | 56271 | 1 | 34 | 0.12 | 0.73 |
| Alprazolam*Stress | 46481 | 1 | 34 | 0.10 | 0.75 |
| Alprazolam*Sex | 153940 | 1 | 34 | 0.33 | 0.57 |
| Stress*Sex | 14920 | 1 | 34 | 0.03 | 0.86 |
| Alprazolam | 19359 | 1 | 34 | 0.04 | 0.84 |
| Stress | 9053 | 1 | 34 | 0.02 | 0.89 |
| Sex | 2104455 | 1 | 34 | 4.53 | **0.04**\* |
| Fig 2I MeA | | | | | |
| Alprazolam*Stress*Sex | 291674 | 1 | 34 | 1.87 | 0.18 |
| Alprazolam*Stress | 4035 | 1 | 34 | 0.03 | 0.87 |
| Alprazolam*Sex | 278969 | 1 | 34 | 1.79 | 0.19 |
| Stress*Sex | 103189 | 1 | 34 | 0.66 | 0.42 |
| Alprazolam | 38692 | 1 | 34 | 0.25 | 0.62 |
| Stress | 6305 | 1 | 34 | 0.04 | 0.84 |
| Sex | 147128 | 1 | 34 | 0.95 | 0.34 |

(*Continued*)

**Table 1.** (Continued)

Fig 3A: **Control group correlations (N = 16; Spearman's coefficient)**

|  |  | PL | CeC | CeL | CeM | ITC | MeA |
|---|---|---|---|---|---|---|---|
| Sociability | r | -0.121 | -0.571 | -0.231 | 0.003 | -0.406 | 0.144 |
|  | p | 0.656 | 0.023* | 0.386 | 0.993 | 0.120 | 0.594 |
| Exploration | r | 0.132 | 0.081 | -0.131 | 0.272 | 0.397 | 0.206 |
|  | p | 0.625 | 0.765 | 0.627 | 0.305 | 0.129 | 0.443 |
| Tail rattling | r | 0.018 | 0.143 | -0.124 | 0.442 | -0.228 | 0.138 |
|  | p | 0.950 | 0.600 | 0.650 | 0.096 | 0.417 | 0.625 |
| Latency to approach | r | 0.153 | 0.407 | -0.165 | 0.427 | -0.032 | 0.445 |
|  | p | 0.570 | 0.118 | 0.538 | 0.100 | 0.908 | 0.086 |

Fig 3B: **Footshock stress group correlations (N = 16; Spearman's coefficient)**

|  |  | PL | CeC | CeL | CeM | ITC | MeA |
|---|---|---|---|---|---|---|---|
| Sociability | r | -0.756 | -0.843 | -0.673 | -0.349 | -0.576 | -0.762 |
|  | p | 0.001** | 0.00008*** | 0.005** | 0.185 | 0.022* | 0.001** |
| Exploration | r | 0.162 | -0.394 | -0.183 | 0.150 | -0.182 | -0.232 |
|  | p | 0.549 | 0.131 | 0.496 | 0.576 | 0.498 | 0.385 |
| Tail rattling | r | -0.175 | -0.122 | -0.268 | 0.046 | -0.273 | -0.200 |
|  | p | 0.513 | 0.649 | 0.313 | 0.865 | 0.303 | 0.455 |
| Latency to approach | r | 0.617 | 0.568 | 0.502 | -0.091 | 0.487 | 0.558 |
|  | p | 0.013* | 0.023* | 0.049* | 0.735 | 0.057 | 0.027* |

## Correlations between cFos levels and behavior following the social approach test

Spearman's correlation coefficient was used to assess the relationship between behavioral variables and cFos expression levels (**Fig 3** and **Table 1**). These correlation analyses were performed for the control and stressed groups. For the brain areas in which we did not find any significant effect of drug x sex, we pooled the data from their respective vehicle or alprazolam treatment groups together for analysis. In controls (**Fig 3A**), there was a significant negative correlation between CeC cFos expression and sociability (r = -0.57, $p$ = 0.023). In the stress group (**Fig 3B**), there was a significant negative correlation between sociability and cFos expression in numerous areas, with more socially active mice demonstrating less neuronal activation in the PL (r = -0.75, $p$ = 0.001), capsular (CeC, r = -0.84, $p$ = 0.00008) and lateral (r = -0.67, $p$ = 0.005) subdivisions of CeA, as well as in the ventromedial ITC (r = -0.58, $p$ = 0.02) and medial amygdala (MeA, r = -0.76, $p$ = 0.001).

Latency to approach the social stimulus positively correlated with the number of cFos+ cells in the PL (r = 0.62, $p$ = 0.01), CeC (r = 0.568, $p$ = 0.023), CeL (r = 0.502, $p$ = 0.050) and MeA (r = 0.55, $p$ = 0.027), with a trend to significance in ITC (r = 0.49, $p$ = 0.06; **Fig 3B**). Distribution of data points and regression lines for the respective brain area for sociability and latency is shown in **Fig 3C and 3D**.

## Discussion

Exposure to footshock is a commonly used model of acute traumatic stress to assay core features of stress disorders such as social withdrawal [30]; however, sociability after footshock stress has been investigated predominantly in male rodents and most other studies that have focused on the effects of stress on social behavior have largely employed chronic stress models which typically lead to social withdrawal [7]. In the current study, several measurements of

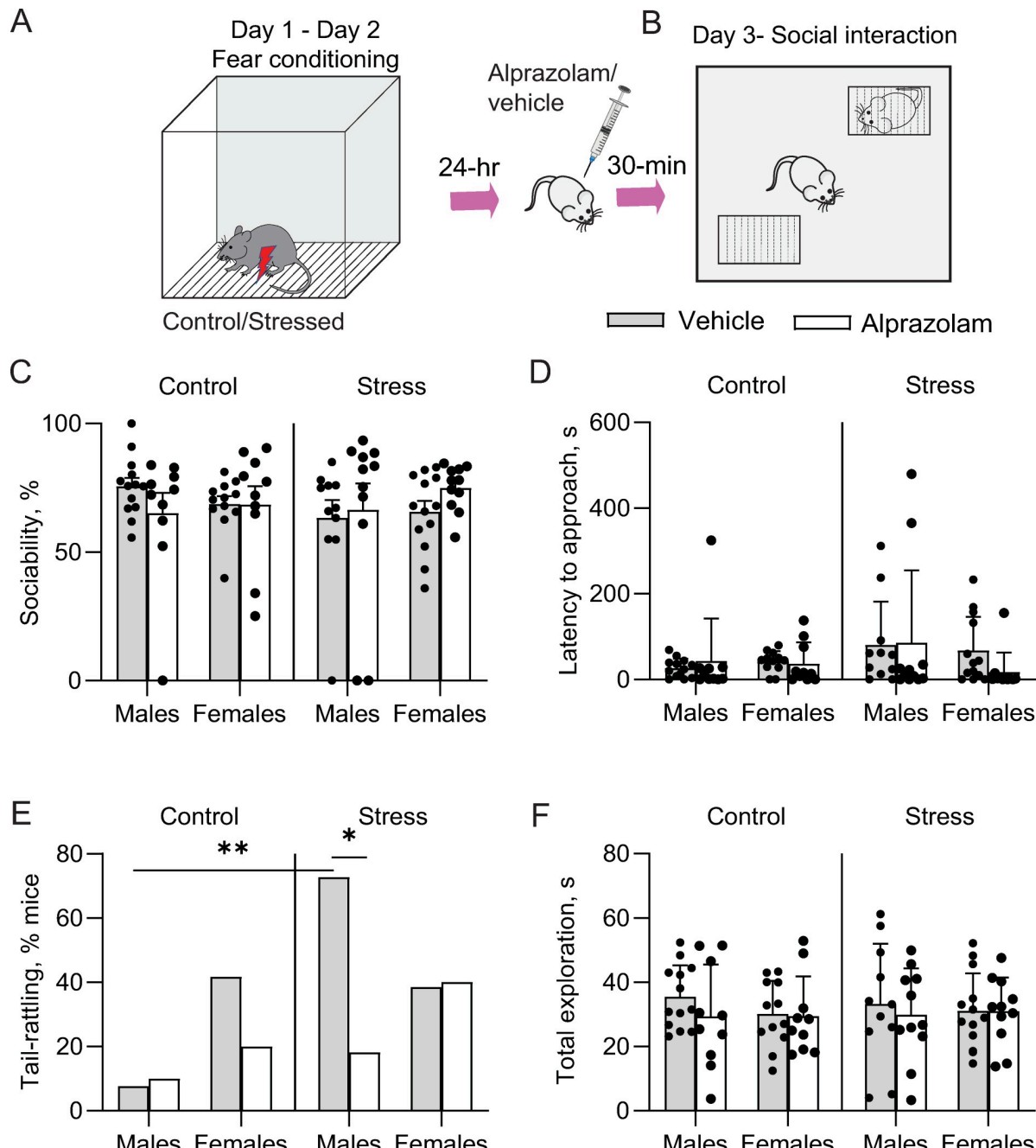

**Fig 1. Effects of acute stress and alprazolam on behavior in the social approach test. A.** Experimental timeline. **B.** Schematic of the social arena. **C.** There was a significant interaction between the drug and stress condition, but there were no significant results from post hoc multiple comparison tests. **D.** Latency to approach was not significantly affected by stress or alprazolam treatment. **E.** The % of male mice show tail rattling behavior increased by stress and was significantly reduced by alprazolam. **F.** Exploratory behavior was not significantly altered by stress or drug treatment. Data are presented as means ± SEM. *$p < 0.05$; *$p < 0.01$ post hoc tests.

social behavior were taken in male and female mice to investigate potential sex differences in the impact of footshock stress on sociability (**Fig 1**) and neuronal activation in the mPFC, CeA, and MeA (**Fig 2**).

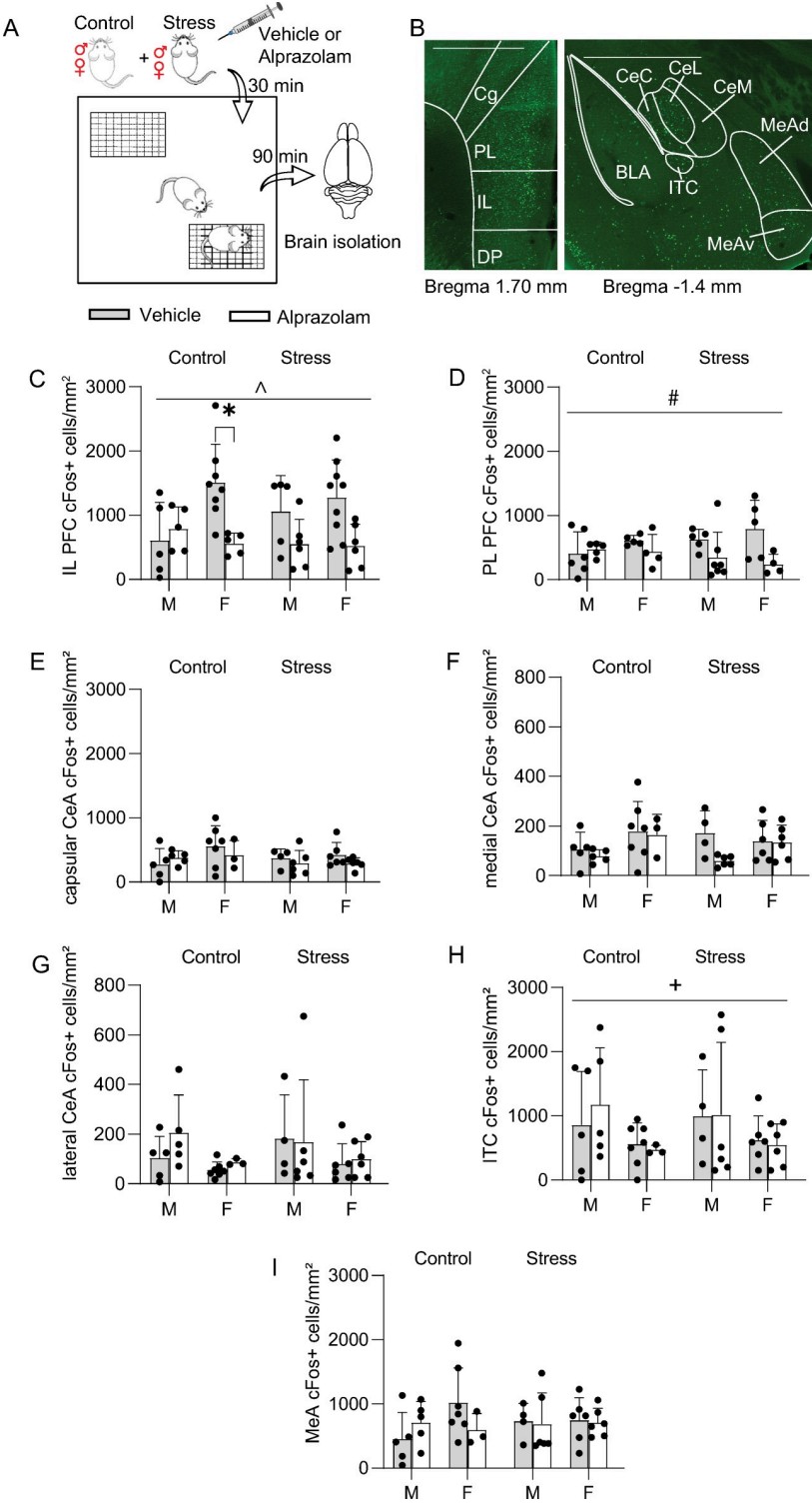

**Fig 2. cFos expression patterns following social approach. A.** Schematic of the experiment. **B.** Representative images of cFos staining results in mPFC (*left*) and amygdala (*right*). CeC = capsular CeA, CeL = lateral CeA, CeM = medial CeA, IL = infralimbic mPFC, ITC = ventromedial intercalated nucleus of amygdala, MeAd = dorsal MeA, MeAv = ventral MeA, PL = prelimbic mPFC. Scale: 1000 μm. **C-I.** Average number of cFos+ cells / mouse. **C.** Alprazolam reduced cFos expression in the IL in control females. **D.** Alprazolam lowered cFos+ cells in the PL. **E-G.** cFos expression in the capsular, medial, or lateral CeA was not significantly affected by stress, sex, or drug. **H.** cFos expression in the ventromedial ITCs were greater in males. **I.** cFos expression in the MeA was unaffected by stress, sex,

or drug. Data are presented as means ± SEM. *$p<0.05$ post hoc tests. ^$p<0.05$, effect of drug x sex; +$p<0.05$, main effect of sex; #$p < 0.05$, main effect of drug.

Although the footshock stress parameters we used did not significantly alter sociability, tail rattling behavior was significantly different between groups during social approach in a sex-dependent manner, being affected by stress and diminished by alprazolam only in males (**Fig 1**). Tail rattling is elicited due to territorial aggression in male mice [31, 32] and it has been suggested that it can be a measurement of threat-induced defensive aggression [26]. Our results are therefore comparable to other mouse sociability studies in which footshock stress produced an enhancement of aggression in male mice [32]. Interestingly, female mice exhibit more tail rattling than males during fear conditioning [26], suggesting that sex differences in tail rattling behavior are both stress- and context-dependent. The significant reduction of tail rattling by alprazolam further supports the link between this behavior and negative valence states. Overall, the observed sex differences reinforce the idea that sex is a crucial factor that should be considered in stress-related studies.

There is evidence that social interaction itself is anxiogenic, even under control conditions [33]. This is supported by our finding of a negative correlation between sociability and cFos expression in the CeA (**Fig 3**). Our results also show that cFos expression in the ITC is higher in males compared to females after social interaction. Given the known role of these amygdala regions in mediating threat responses [21, 34], these data suggest that social interaction may induce a higher level of defensiveness in male mice. Consistent with this hypothesis, previous work has shown that CeA neuronal activation is associated with male mouse aggression during social interaction in the resident intruder assay [18, 35]. These findings emphasize the need to study sex differences while deciphering the relationship between stress and social behaviors. Further investigations should be conducted into sex differences in different neuronal populations of CeA, expressing molecular markers such as somatostatin or corticotropin-releasing factor. This is especially relevant given that there are more corticotropin-releasing factor receptor 1-containing neurons in the male CeA compared to female, and they have different neuronal excitability in response to corticotropin-releasing factor [36].

cFos expression in PL and IL mPFC in our study was not significantly affected by prior footshock stress exposure; however, PL and IL cFos levels were decreased due to alprazolam treatment both in stressed and control mice, which is consistent with the observed decrease in mPFC activation in humans given benzodiazepines [37]. Acute restraint stress has been shown to activate PL, but not IL or CeA, and social interaction following stress enhances PL activation without affecting CeA activation [38]. The cFos expression patterns observed in our study may be due to the delay (at least 24 hr.) between stress exposure and sacrifice of the animals.

Among control vehicle-injected mice, females had more cFos+ cells in IL. Tan et al. obtained comparable results in the mPFC of females after chronic adolescent social isolation stress, with pyramidal neurons being less activated during sociability tests [39]. Alprazolam significantly lowered the number of cFos+ cells in the IL compared to vehicle-injected female controls, consistent with the benzodiazepine effects observed in humans: women demonstrate decreased activity in frontal regions after treatment, while an opposite effect is present in males [40]. This effect could be related to females having significantly higher GABA-A benzodiazepine receptor availability [41].

The MeA has been linked to a wide variety of social behaviors, such as aggression, mating, and parenting [42]. Here, we find that MeA cFos expression induced by the social approach test is not affected by sex, stress, or alprazolam. Prior studies have demonstrated that socially defeated females housed with aggressive male residents exhibit increased cFos activation in the

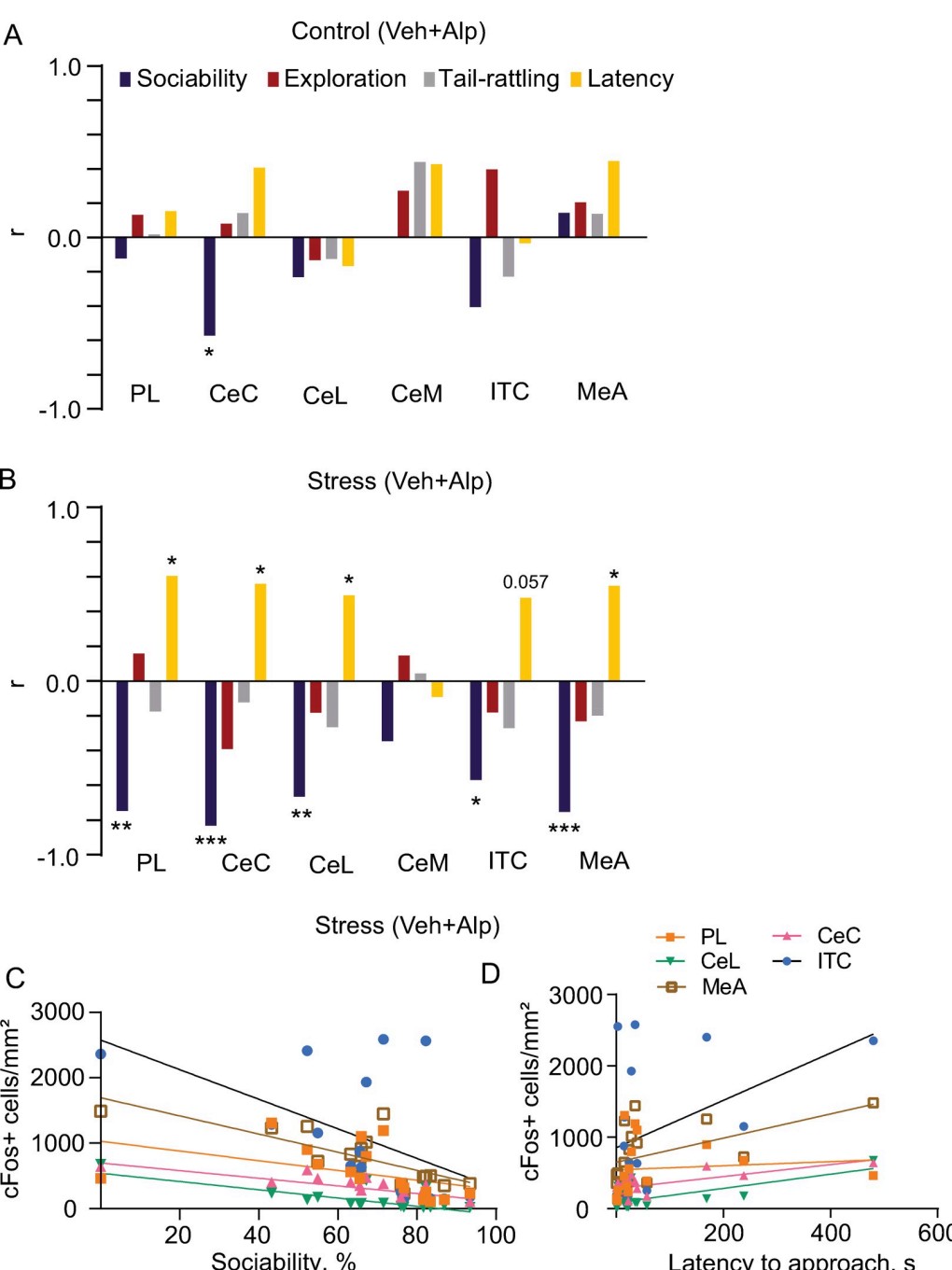

**Fig 3. Correlations between behavioral variables and cFos expression.** Spearman's correlation coefficients (r) between the number of cFos+ cells and behavioral parameters in PL, CeA subdivisions, ITC and MeA of vehicle (Veh) or alprazolam (Alp) treated mice pooled together in the control (**A; N = 16**) and stressed (**B; N = 16**) groups. **C, D,** the distribution of correlation points for sociability and latency to approach against cFos+ cells/mm². Spearman's correlation coefficient represented with * indicating significant correlations (*$p < 0.05$, **$p < 0.001$; ***$p < 0.0001$). CeC = capsular CeA, CeM = medial CeA, CeL = lateral CeA, ITC = ventromedial intercalated nucleus of amygdala, MeA = medial nucleus of the amygdala, PL = prelimbic mPFC.

MeA [43], and the MeA is more responsive to aggressive than to benign social interaction [44]. We designed our experiments to exclude the possibility of aggressive interactions, therefore

free interactions between the experimental and stimulus mice were not possible. Future studies could investigate the effects of stress and sex on MeA activation in free interaction paradigms.

In stressed animals, sociability was negatively correlated with cFos expression in the PL, CeC and CeL, as well as in the ventromedial ITC and MeA. In non-stressed animals, MeA activation has been demonstrated previously because of social interaction [45], and an opposite relationship has been shown between CeA cFos expression and sociability after anxiogenic synthetic amphetamine treatment [46]. Because mPFC activation has been demonstrated to suppress social behaviors [47], the negative correlation observed in the current study could be expected, although enhanced activity of a subset of mPFC neurons was correlated with social approach behavior previously [48]. In controls, there was a significant negative correlation between sociability and CeC cFos expression, and an opposite correlation between the latter and the latency to approach. Latency to approach the social stimulus also positively correlated with the number of cFos+ cells in the PL and MeA.

In conclusion, we found that two days of footshock exposure at levels commonly used in fear conditioning did not elicit changes in mouse sociability. However, we did find sex differences in defensive tail rattling, the effects of alprazolam on defensive tail rattling, and cFos expression during the social approach assay. The social consequences of stress have been extensively studied using the social defeat stress model that entails exposure to emotional / psychological stress and leads to depression-related outcomes [7]. After the social defeat procedure, most mice develop a decreased drive to approach and interact with the social target [49]; however, social defeat stress is difficult to achieve in female mice. Acute stress paradigms utilizing footshock could facilitate investigations into sex differences in social behavior following trauma, which would be valuable in the search for sex-specific mechanisms involved in the pathophysiology of post-traumatic stress disorder (PTSD) and could facilitate the development of personalized therapeutic interventions. Future work should therefore define the optimal conditions, such as footshock intensity or lighting conditions, that influence sociability after stress. It could be especially valuable to develop paradigms that stratify mice as resilient and susceptible to further validate acute footshock stress as a tool for PTSD research.

## Author Contributions

**Conceptualization:** Mariia Dorofeikova, Chandrashekhar D. Borkar, Jonathan P. Fadok.

**Formal analysis:** Mariia Dorofeikova, Chandrashekhar D. Borkar, Jonathan P. Fadok.

**Funding acquisition:** Jonathan P. Fadok.

**Investigation:** Mariia Dorofeikova, Chandrashekhar D. Borkar, Katherine Weissmuller, LydiaSmith-Osborne, Samhita Basavanhalli, Erin Bean, Avery Smith, Anh Duong, Alexis Resendez.

**Methodology:** Mariia Dorofeikova, Chandrashekhar D. Borkar, Jonathan P. Fadok.

**Project administration:** Mariia Dorofeikova, Jonathan P. Fadok.

**Resources:** Jonathan P. Fadok.

**Visualization:** Mariia Dorofeikova, Chandrashekhar D. Borkar, Jonathan P. Fadok.

**Writing – original draft:** Mariia Dorofeikova, Jonathan P. Fadok.

**Writing – review & editing:** Mariia Dorofeikova, Chandrashekhar D. Borkar, Katherine Weissmuller, Lydia Smith-Osborne, Jonathan P. Fadok.

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
