## [Decision Letter · Decision Letter 0]

26 Aug 2022

PONE-D-22-19470Effects of footshock stress on social behavior and neuronal activation in the medial prefrontal cortex and amygdala of male and female micePLOS ONE

Dear Dr. Fadok,

Thank you for submitting your manuscript to PLOS ONE. After careful consideration, we feel that it has merit but does not fully meet PLOS ONE’s publication criteria as it currently stands. Therefore, we invite you to submit a revised version of the manuscript that addresses the points raised during the review process.

We look forward to receiving your revised manuscript.

Kind regards,

Alexandra Kavushansky, PhD

Academic Editor

PLOS ONE

Journal Requirements:

"This work was supported by the Louisiana Board of Regents through the Board of Regents Support Fund (LEQSF(2018-21)-RD-A-17) and the National Institute of Mental Health of the National Institutes of Health under award number R01MH122561 to JPF."

"This work was supported by the Louisiana Board of Regents through the Board of Regents Support Fund (LEQSF(2018-21)-RD-A-17) and the National Institute of Mental Health of the National Institutes of Health under award number R01MH122561 to JPF. The content is solely the responsibility of the authors and does not necessarily represent the official views of the National Institutes of Health."

Reviewers' comments:

Reviewer's Responses to Questions

**Comments to the Author**

1. Is the manuscript technically sound, and do the data support the conclusions?

Reviewer #1: Yes

Reviewer #2: No

Reviewer #3: Partly

2. Has the statistical analysis been performed appropriately and rigorously? 

Reviewer #1: Yes

Reviewer #2: Yes

Reviewer #3: No

3. Have the authors made all data underlying the findings in their manuscript fully available?

Reviewer #1: Yes

Reviewer #2: Yes

Reviewer #3: Yes

4. Is the manuscript presented in an intelligible fashion and written in standard English?

Reviewer #1: Yes

Reviewer #2: Yes

Reviewer #3: Yes

5. Review Comments to the Author

Reviewer #1: 1. Main theme in summary

Authors aimed to demonstrate the sex-dependent effects of acute traumatic stress on mouse sociability using two-days of footshock stress followed by social interaction test, and its correlation with cFos expression in the medial prefrontal cortex and amygdala. The reversibility effects of acute benzodiazepine alprozalam treatment on these behavioral and neuronal changes were also investigated. Study findings do offer a potential correlation of sex differences in defensive behavior in male mice, but not sociability, and the underlying cFos activation, as well as the reversibility effects with acute alprazolam treatment.

2. Strengths

Overall, this is quite an interesting study which has generated good results to describe sex differences in the effects acute traumatic stress on social behaviors and the underlying neural substrates involvement.

3. Limitation

The study design (i.e., the groups allocation and treatment) are unclear. Require elaboration and clarification - elaborated further in methods section.

4. Detailed remarks

Abstract - needs revision for the following:

• The problem statement cannot be clearly seen from the abstract.

• Authors should add number of animals (i.e., N total, N for each group), the route of alprazolam administration, p-value

• Authors should also highlight on the reversal effects of alprazolam treatment in abstract conclusion.

Introduction is satisfactory overall, with additional info necessary to improve clarity to the research background as below:

• Para 2, Line 47: “One common model of traumatic stress in rodents is footshock exposure” – Authors should briefly describe previous studies using footshock exposure for social behaviors

• Para 5: Authors should briefly describe on benzodiazepine alprozalam and its common indication (e.g: anxiolytic? Antidepressant effects?)

• Para 5, Line 74,75: “Therefore, we performed social interaction tests 24 hours after two consecutive days of footshock stress” – More suitable to be in the Methodology sections and therefore to be removed from Introduction.

Materials and Methods - some missing info needed to be updated, and additional clarity as follow:

• Ethics approval protocol number should be added in the text.

• Line 87-88: “Unfamiliar strain-, sex- and age-matched mice were used as the passively interacting counterparts (stimulus mice) during social interaction tests.” – Author should disclose the number of stimulus mice used for this study.

• The assignment of groups and treatment are unclear. I suggest to include a Grouping subsection in the Methodology, instead of in the Results section. Summarised on the male vs female, footshock vs control context exposure, vehicle vs alprozalam group allocations, and the total number of animals (i.e., N total, N for each group). It would be helpful to include one figure that summarised the overall study design (i.e., the treatment groups, schedules up to whole brain collection) for better clarity of the group allocations.

• Is the behavioral study design (ie., duration, intensity of footshock exposure, design of social interaction test etc) based on previous protocols? If yes, please cite the previous study(s).

Results - Overall is satisfactory and met study objectives with one minor remark as follow:

• The quality of Figure 1C-F can be further improved – in present form, hard to read and appreciate the results.

Discussion - overall is satisfactory with some clarifications/additions suggested below:

• Para 3, Line 280-281 “…which mostly employed chronic stress models.” – should briefly describe the effects of chronic stress on social behaviour from previous animal/rodent studies.

• Para 5, Line 302-304: “…which may occur due to the delay between stress exposure and sacrifice of the animals.” – What is the estimated duration between stress exposure and animal sacrifice in your study? Is it based on previous protocol? Suggestion of duration for future studies?

• Concluding paragraph: Authors should briefly describe overall findings from this study, highlighting on the potential sex differences in stress-related studies. Emphasise on the need to understand these mechanisms in both sexes, which will provide new insights into the sex dimorphism documented in the pathophysiology of PTSD and possibly help facilitate the development of sex-specific therapeutic interventions.

Reviewer #2: Male and female mice underwent footshock stress for 2 days and were then exposed to a novel stimulus mouse. Half of the mice received alprazolam and half received vehicle. cFos protein expression was assessed in subregions of the prefrontal cortex and amygdala. No effects of stress were observed on social approach, but tail rattling was increased somewhat in males. There were some effects of sex and alprazolam on cFos but no straightforward main effects or interactions involving stress.

The main concern with this experiment is that five 1 sec footshocks appear insufficiently stressful to elicit long-lasting changes in social interaction behavior.

No effects of stress on social interaction were observed in males or females, but there was increased tail rattling in males exposed to footshock stress. Why should we care about the effects of stress on cFos expression when there are no robust behavioral effects on social interaction per se, or on latency to interact. Similarly the effects of alprazolam would be more interesting if there were robust behavioral effects of stressor exposure using these procedures.

Other minor concerns are listed below.

Why were the experimental mice singly housed for 7 days? There is no rationale provided for this.

Mice were exposed to a conspecific in a mesh box and there was no physical interaction, so it may better be described as social approach. Since there was no actual interaction.

Experiments were done during the light phase, the inactive phase in rodents. Time of day has profound effects on behavior, and results may be more clinically relevant when experiments are done during the active phase.

As noted by the authors on line 341, “Future work should define the optimal conditions, such as footshock intensity or lighting conditions, that influence sociability after stress.” Those conditions should have been defined using pilot studies before this experiment was performed.

Reviewer #3: Overall, this is a straightforward experiment and the research question was clear - to determine whether acute stress (2 days of inescapable footshock) changed social behavior in a sex-dependent manner and whether these effects were reversible with the anxiolytic, alprazolam. The manuscript is well written, and overall find the authors to have sufficiently summarized and integrated the existing literature into their intro and discussion. With that said, I have some issues with the study design, statistics and reporting of significant effects which are enumerated below:

1. The authors state that "male (N = 18 control, N = 22 stress) and female (N = 17 control, N = 24 stress) mice were allotted to the vehicle (N = 49, 24 males and 25 females) and alprazolam (N = 32, 16 males and 16 females) treatment groups" but do not explicitly state how many vehicle were from stress or control conditions, i.e., female-control-vehicle = x, male-control-vehicle = y, so it is clear how the groups are balanced. Best I can tell (but should not have to guess or count, please spell this out) there are:

Male-Control-Vehicle = 13

Male-Stress-Vehicle = 11

Male-Control-Alprazolam = 5

Male-Stress-Alprazolam = 11

Female-Control-Vehicle = 12

Female-Stress-Vehicle = 13

Female-Control-Alprazolam = 5

Female-Stress-Alprazolam = 11

Thus it appears that the groups are very unbalanced, and statistically this is problematic because of the inherent variability this introduces, and the likelihood that this is violating the equal variances assumptions for all the ANOVAs performed. For example, the authors report a significant interaction between stress and alprazolam, which is clearly being driven by 3 extreme cases across the male and females in the control condition where they have grossly under-sampled and thus extreme data points have large influence. The authors need to increase the animal numbers in their control-alprazolam conditions in order to make this data interpretable. This applies to every statistical comparison made in this manuscript.

2. For the tail rattling analysis, the Fisher's exact test is, from my understanding, a test that can explore a 2 x 2 interaction, not the three way they have here. So this test may not be the most appropriate. However, probability stats should be reported along with p values at a minimum, along with a clear explanation of which factors were compared that produced such probability values.

3. Related to comment 1, when we get to the cFos analysis, group numbers change with no explanation as to how they selected the animals they analyzed:

Male-Control-Vehicle = 13 -> 5

Male-Stress-Vehicle = 11 -> 5

Male-Control-Alprazolam = 5 -> 5

Male-Stress-Alprazolam = 11 -> 6

Female-Control-Vehicle = 12 -> 8

Female-Stress-Vehicle = 13 -> 9

Female-Control-Alprazolam = 5 -> 5

Female-Stress-Alprazolam = 11 -> 6

This is less unbalanced, but without a clear explanation of how animals were selected, the process could have been purposefully or accidentally biased in some manner, and may largely skew the correlational analyses.

3. The authors include animals in their cFos graphs that seemingly did not go through stress or social interaction procedures. These appear to not be included in any analyses or methodological description and their inclusion should be described and either properly included in the analyses or excluded from the manuscript.

4. In the correlational analysis for cFos against sociability and latency to approach, it is unclear if the same animals are used across all of the brain regions. For example, if all the same animals were used for the cFos analysis you would have cFos data counts in sets of 5 for each sociability score, which you can see often, as in Fig 3C, data points for each brain region hovering at ~94% sociability you can clearly see one set of 5, but if you look just to the left at ~90%, there is only one orange square at this sociability score, so I am just trying to understand how these data were derived and how animals were included for some brain regions and not others, some animals with extreme scores were included and other excluded from this analysis. This occurs in the latency graph (3D) as well.

Overall, the authors are not making any grandiose conclusions from their data, but I believe more controls need to be run and more careful attention paid to statistical analyses. As well, explicit criteria for inclusion or exclusion in subsequent cFos analyses should be given.

6. PLOS authors have the option to publish the peer review history of their article (what does this mean?). If published, this will include your full peer review and any attached files.

Reviewer #1: **Yes: **Muzaimi Mustapha

Reviewer #2: No

Reviewer #3: No

---

## [Author Response · Author response to Decision Letter 0]

20 Dec 2022

Comments to the Author

1. Is the manuscript technically sound, and do the data support the conclusions?

Reviewer #1: Yes

Reviewer #2: No

Reviewer #3: Partly

2. Has the statistical analysis been performed appropriately and rigorously? 

Reviewer #1: Yes

Reviewer #2: Yes

Reviewer #3: No

3. Have the authors made all data underlying the findings in their manuscript fully available?

Reviewer #1: Yes

Reviewer #2: Yes

Reviewer #3: Yes

4. Is the manuscript presented in an intelligible fashion and written in standard English?

Reviewer #1: Yes

Reviewer #2: Yes

Reviewer #3: Yes

5. Review Comments to the Author

We thank the reviewers for their thoughtful critiques of our manuscript. We have addressed all the reviewer’s comments with additional experiments, data analysis, statistical testing, and major revisions to the text. We feel the revised manuscript is improved. 

Our point-by-point responses are in bold text below and the changes made to the manuscript are highlighted in the Revised Manuscript with Track Changes file.

Reviewer #1: 1. Main theme in summary

Authors aimed to demonstrate the sex-dependent effects of acute traumatic stress on mouse sociability using two-days of footshock stress followed by social interaction test, and its correlation with cFos expression in the medial prefrontal cortex and amygdala. The reversibility effects of acute benzodiazepine alprozalam treatment on these behavioral and neuronal changes were also investigated. Study findings do offer a potential correlation of sex differences in defensive behavior in male mice, but not sociability, and the underlying cFos activation, as well as the reversibility effects with acute alprazolam treatment.

2. Strengths

Overall, this is quite an interesting study which has generated good results to describe sex differences in the effects acute traumatic stress on social behaviors and the underlying neural substrates involvement.

We thank the reviewer for their positive feedback on our manuscript.

3. Limitation

The study design (i.e., the groups allocation and treatment) are unclear. Require elaboration and clarification - elaborated further in methods section.

We have added clear information about group allocation and treatment in the revised manuscript under the Groups subheading in the Methods.

4. Detailed remarks

Abstract - needs revision for the following:

• The problem statement cannot be clearly seen from the abstract.

We clarified the intention of the study by stating in the Abstract, “The aim of this study was to explore the effect of footshock stress, using analogous parameters to those commonly used in fear conditioning assays, on the sociability of male and female C57Bl/6J mice in a standard social interaction test.”

• Authors should add number of animals (i.e., N total, N for each group), the route of alprazolam administration, p-value

We have added the total number of animals as well as the group composition to the abstract. We also added the route of drug administration (i.p.), and the p-values for the significant mentioned results.

• Authors should also highlight on the reversal effects of alprazolam treatment in abstract conclusion.

We have highlighted the effects of alprazolam by stating “This stress-induced increase in tail-rattling was alleviated by alprazolam (p = 0.03), yet alprazolam had no effect on female tail-rattling behavior in the stress group. Alprazolam lowered cFos expression in the medial prefrontal cortex (p = 0.001 infralimbic area, p = 0.02 prelimbic area), and social interaction induced sex-dependent differences in cFos activation in the ventromedial intercalated cell clusters (p = 0.04).”

Introduction is satisfactory overall, with additional info necessary to improve clarity to the research background as below:

• Para 2, Line 47: “One common model of traumatic stress in rodents is footshock exposure” – Authors should briefly describe previous studies using footshock exposure for social behaviors

We have added the previous literature on footshock exposure induced traumatic stress as follows.

“In rats, footshock stress has been shown to induce social avoidance in animals with elevated levels of fear generalization [Dong et al., 2020] and impair the response to the social-paired compartment which was reversed by antidepressant treatment [Daniels et al. 2021]. Moreover, intense footshock (2 mA, 10 s) followed by three weekly situational reminders elicit long-term impairments in social interaction in female rats [Louvart et al., 2005]. ”

• Para 5: Authors should briefly describe on benzodiazepine alprozalam and its common indication (e.g: anxiolytic? Antidepressant effects?)

We have specified that alprazolam is used in short-term management of anxiety disorders.

• Para 5, Line 74,75: “Therefore, we performed social interaction tests 24 hours after two consecutive days of footshock stress” – More suitable to be in the Methodology sections and therefore to be removed from Introduction.

We have modified the relevant statement as “We hypothesized that two days of footshock stress would negatively affect sociability, and that those changes might depend on sex and be reversible with the fast-acting benzodiazepine alprazolam, that which is used in short-term management of anxiety disorders.” 

Materials and Methods - some missing info needed to be updated, and additional clarity as follow:

• Ethics approval protocol number should be added in the text.

We have added the IACUC ethical approval protocol number (Protocol ID-1013).

• Line 87-88: “Unfamiliar strain-, sex- and age-matched mice were used as the passively interacting counterparts (stimulus mice) during social interaction tests.” – Author should disclose the number of stimulus mice used for this study.

 We have used a total of 33 mice as the stimulus mice. This information is included in the revised version of the manuscript.

• The assignment of groups and treatment are unclear. I suggest to include a Grouping subsection in the Methodology, instead of in the Results section. Summarised on the male vs female, footshock vs control context exposure, vehicle vs alprozalam group allocations, and the total number of animals (i.e., N total, N for each group). It would be helpful to include one figure that summarised the overall study design (i.e., the treatment groups, schedules up to whole brain collection) for better clarity of the group allocations.

We have added the following information in the revised manuscript. 

A total of 45 males and 46 females were separated into the following groups: 

Control males treated with vehicle, N = 13, 

Control females treated with vehicle, N = 12, 

Control males treated with alprazolam, N = 10, 

Control females treated with alprazolam, N = 10, 

Stressed males treated with vehicle, N = 11, 

Stressed females treated with vehicle, N = 13, 

Stressed males treated with alprazolam, N = 11, 

Stressed females treated with alprazolam, N = 11.

• Is the behavioral study design (ie., duration, intensity of footshock exposure, design of social interaction test etc) based on previous protocols? If yes, please cite the previous study(s).

We used footshock duration and intensity based on our previous study [Borkar et al. 2020] and others [Bali and Jaggi, 2015]. We employed the method of the social interaction scoring from Macbeth et al., 2009 [27]. We have incorporated this information in the manuscript. 

Results - Overall is satisfactory and met study objectives with one minor remark as follow:

• The quality of Figure 1C-F can be further improved – in present form, hard to read and appreciate the results.

As suggested, we have revised Figure 1 to be of higher quality. 

Discussion - overall is satisfactory with some clarifications/additions suggested below:

• Para 3, Line 280-281 “…which mostly employed chronic stress models.” – should briefly describe the effects of chronic stress on social behaviour from previous animal/rodent studies.

We have rewritten the Discussion to convey a clearer message overall. We have added the following clarification. “most other studies that have focused on the effects of stress on social behavior have largely employed chronic stress models which typically lead to social withdrawal [Toth and Neumann, 2013; Louvart et al., 2005].”

• Para 5, Line 302-304: “…which may occur due to the delay between stress exposure and sacrifice of the animals.” – What is the estimated duration between stress exposure and animal sacrifice in your study? Is it based on previous protocol? Suggestion of duration for future studies?

There was ~ 48 hours delay from the first footshock session and ~ 24 hours delay from the last footshock session to the experimental endpoint of the study. We have clarified this in the Discussion. Our overall goal was to assay cFos following social interaction (as opposed to stress itself) in the separate groups of animals. Therefore, we sacrificed subjects 90 minutes following social interaction, following common convention in the field. (For example, Zhong, J. et al., 2014, https://doi.org/10.1186/s13041-014-0066-x). 

In addition, we have added the following statements regarding future directions. “Future work should define the optimal conditions, such as footshock intensity or lighting conditions, that influence sociability after stress. It could be especially valuable to develop paradigms that stratify mice as resilient and susceptible to further validate acute footshock stress as a tool for PTSD research.”

• Concluding paragraph: Authors should briefly describe overall findings from this study, highlighting on the potential sex differences in stress-related studies. Emphasize on the need to understand these mechanisms in both sexes, which will provide new insights into the sex dimorphism documented in the pathophysiology of PTSD and possibly help facilitate the development of sex-specific therapeutic interventions.

As suggested, we summarize the main findings in the concluding paragraph, and we have added the following statement. “Acute stress paradigms utilizing footshock facilitate investigations into sex differences in social behavior following trauma, which would be valuable for the search of sex-specific mechanisms involved in the pathophysiology post-traumatic stress disorder (PTSD) and could facilitate the development of personalized therapeutic interventions.”

Reviewer #2: Male and female mice underwent footshock stress for 2 days and were then exposed to a novel stimulus mouse. Half of the mice received alprazolam and half received vehicle. cFos protein expression was assessed in subregions of the prefrontal cortex and amygdala. No effects of stress were observed on social approach, but tail rattling was increased somewhat in males. There were some effects of sex and alprazolam on cFos but no straightforward main effects or interactions involving stress.

The main concern with this experiment is that five 1 sec footshocks appear insufficiently stressful to elicit long-lasting changes in social interaction behavior.

The aim of this study was to elucidate the effects of two days of acute footshock stress on behavior during the social interaction assay, using shock parameters commonly used in Pavlovian fear conditioning studies. We use routinely use these parameters (0.9 mA, 1 s duration) to elicit a complex array of defensive responses to threat (see Fadok et al. 2017, Borkar et al. 2020 for examples). Because these shock parameters induce robust escape behavior, including jumping, we hypothesized that they would be strong enough to induce changes in social interaction. Although our data did not support this hypothesis, we believe it is important to publish these negative data as they provide vital information for others in the community who may be interested in conducting similar studies. The omission of negative data in neuroscience publications is a fundamental problem that can lead to wasted resources, time, and animal life. Additionally, although we did not see a difference in the shock-exposed group in social interaction or latency to approach, we did find sex-dependent differences in defensive tail rattling behavior, as well as sex-dependent differences in the effect of alprazolam on tail rattling behavior. These data will be of interest to researchers interested in social valence and sex differences.

No effects of stress on social interaction were observed in males or females, but there was increased tail rattling in males exposed to footshock stress. Why should we care about the effects of stress on cFos expression when there are no robust behavioral effects on social interaction per se, or on latency to interact. Similarly the effects of alprazolam would be more interesting if there were robust behavioral effects of stressor exposure using these procedures.

We present the cFos data because it provides the reader with an assessment of the extent to which neuronal activation in subdivisions of the medial prefrontal cortex and amygdala is impacted by social interaction, previous stress exposure, sex, and alprazolam. We find that alprazolam reduces cFos in the PL and IL regions of the medial prefrontal cortex, and there is a significant reduction in cFos in the IL of control females. We also found a significant effect of sex on cFos expression in the ventromedial intercalated nucleus of amygdala, with greater expression in males. These sex-dependent effects will be of potential interest to researchers interested in sex differences in brain function and the effects of drugs on brain activation.

Other minor concerns are listed below.

Why were the experimental mice singly housed for 7 days? There is no rationale provided for this.

We used single housing to avoid potential aggression in male mice exposed to footshock. This is customary practice in our laboratory during fear conditioning studies. Moreover, single housing for 1 week is a relatively short duration and may not be as stressful as longer durations of social isolation (e.g. 5-11 weeks; Buckinx et al., 2021, PMID: 34776890). We have added this justification in the Methods section. 

Mice were exposed to a conspecific in a mesh box and there was no physical interaction, so it may better be described as social approach. Since there was no actual interaction.

Although the mice were able to freely sniff one another, we agree that full physical interaction was not possible. We have changed the language to reflect this through the manuscript.

Experiments were done during the light phase, the inactive phase in rodents. Time of day has profound effects on behavior, and results may be more clinically relevant when experiments are done during the active phase.

We agree that time of day has a profound impact on behavior. Because we were interested in observing the effects of shock based on fear conditioning parameters, we performed our experiments during the light phase, which is when our lab and many others perform fear conditioning studies.

As noted by the authors on line 341, “Future work should define the optimal conditions, such as footshock intensity or lighting conditions, that influence sociability after stress.” Those conditions should have been defined using pilot studies before this experiment was performed.

We have modified the last two paragraphs of the Discussion to clarify the intention of our study and to emphasize that although the parameters we used did not affect social approach, future studies should determine the optimal conditions because of the potential translational relevance.

Reviewer #3: Overall, this is a straightforward experiment and the research question was clear - to determine whether acute stress (2 days of inescapable footshock) changed social behavior in a sex-dependent manner and whether these effects were reversible with the anxiolytic, alprazolam. The manuscript is well written, and overall find the authors to have sufficiently summarized and integrated the existing literature into their intro and discussion. With that said, I have some issues with the study design, statistics and reporting of significant effects which are enumerated below:

1. The authors state that "male (N = 18 control, N = 22 stress) and female (N = 17 control, N = 24 stress) mice were allotted to the vehicle (N = 49, 24 males and 25 females) and alprazolam (N = 32, 16 males and 16 females) treatment groups" but do not explicitly state how many vehicle were from stress or control conditions, i.e., female-control-vehicle = x, male-control-vehicle = y, so it is clear how the groups are balanced. Best I can tell (but should not have to guess or count, please spell this out) there are:

Male-Control-Vehicle = 13

Male-Stress-Vehicle = 11

Male-Control-Alprazolam = 5

Male-Stress-Alprazolam = 11

Female-Control-Vehicle = 12

Female-Stress-Vehicle = 13

Female-Control-Alprazolam = 5

Female-Stress-Alprazolam = 11

Thus it appears that the groups are very unbalanced, and statistically this is problematic because of the inherent variability this introduces, and the likelihood that this is violating the equal variances assumptions for all the ANOVAs performed. For example, the authors report a significant interaction between stress and alprazolam, which is clearly being driven by 3 extreme cases across the male and females in the control condition where they have grossly under-sampled and thus extreme data points have large influence. The authors need to increase the animal numbers in their control-alprazolam conditions in order to make this data interpretable. This applies to every statistical comparison made in this manuscript.

We thank the reviewer for their positive feedback and careful review of our manuscript. We appreciate the reviewer’s concerns about group balance, statistics, and better reporting of group composition. To address this concern, we added 5 mice to the male-control-alprazolam group and 5 mice to the female-control-alprazolam group. We then retested all the data using ANOVAs. There are no significant differences in sociability, latency to approach, or exploration time.

We have added the following group breakdown to the Methods section:

“Groups 

A total of 45 males and 46 females were separated into the following groups: 

Control males treated with vehicle, N = 13, 

Control females treated with vehicle, N = 12, 

Control males treated with alprazolam, N = 10, 

Control females treated with alprazolam, N = 10, 

Stressed males treated with vehicle, N = 11, 

Stressed females treated with vehicle, N = 13, 

Stressed males treated with alprazolam, N = 11, 

Stressed females treated with alprazolam, N = 11.”

2. For the tail rattling analysis, the Fisher's exact test is, from my understanding, a test that can explore a 2 x 2 interaction, not the three way they have here. So this test may not be the most appropriate. However, probability stats should be reported along with p values at a minimum, along with a clear explanation of which factors were compared that produced such probability values.

We agree that the Fisher’s exact test is designed to test 2 x 2 contingency tables and not three-way comparisons. This was the way we tested the data (between group comparisons with rattle versus no rattle), but we presented the data as counts in the figure. We realize that this was not the best way to present the data in the figure. We have changed the panel to reflect that we tested the contingencies using the proportion of mice in each group that tail rattled or not. 

3. Related to comment 1, when we get to the cFos analysis, group numbers change with no explanation as to how they selected the animals they analyzed:

Male-Control-Vehicle = 13 -> 5

Male-Stress-Vehicle = 11 -> 5

Male-Control-Alprazolam = 5 -> 5

Male-Stress-Alprazolam = 11 -> 6

Female-Control-Vehicle = 12 -> 8

Female-Stress-Vehicle = 13 -> 9

Female-Control-Alprazolam = 5 -> 5

Female-Stress-Alprazolam = 11 -> 6

This is less unbalanced, but without a clear explanation of how animals were selected, the process could have been purposefully or accidentally biased in some manner, and may largely skew the correlational analyses.

We apologize for not clearly stating how mice were selected for the cFos analysis. All included mice were randomly selected from the separate groups for histological analysis. All tissue processing, imaging, and analysis was performed by investigators blinded to experimental conditions. We have ensured that this information is included in the Methods.

3. The authors include animals in their cFos graphs that seemingly did not go through stress or social interaction procedures. These appear to not be included in any analyses or methodological description and their inclusion should be described and either properly included in the analyses or excluded from the manuscript.

The “No SI” group was a home cage control and was included to serve as a qualitative measurement of cFos changes in the different brain areas. Because there were only 2 males and 3 females in the No SI group, we did not include these data in the quantitative analysis. We apologize for not describing this group in the methods. We have excluded these data from the revised manuscript.

4. In the correlational analysis for cFos against sociability and latency to approach, it is unclear if the same animals are used across all of the brain regions. For example, if all the same animals were used for the cFos analysis you would have cFos data counts in sets of 5 for each sociability score, which you can see often, as in Fig 3C, data points for each brain region hovering at ~94% sociability you can clearly see one set of 5, but if you look just to the left at ~90%, there is only one orange square at this sociability score, so I am just trying to understand how these data were derived and how animals were included for some brain regions and not others, some animals with extreme scores were included and other excluded from this analysis. This occurs in the latency graph (3D) as well.

We thank the reviewer for highlighting the issue regarding the variable number of animals included in the correlation analysis. We have revised Figure 3 considering this. In the revised version we have used cFos and behavioral data from 8 males and 8 females randomly selected from the Alp and Veh treated groups (which have cFos data for all brain nuclei) to analyze the correlations in mice subjected to the stress and control group. The data points represent the correlation of sociability or latency to approach in 5 brain regions from the respective animal. 

Overall, the authors are not making any grandiose conclusions from their data, but I believe more controls need to be run and more careful attention paid to statistical analyses. As well, explicit criteria for inclusion or exclusion in subsequent cFos analyses should be given.

As suggested, we added 5 males and 5 females in the control+Alprazolam treatment group to balance the group composition and we retested the data statistically. In addition, we have clarified the inclusion criteria used for the selection of mice for cFos analyses.

---

## [Decision Letter · Decision Letter 1]

23 Jan 2023

Effects of footshock stress on social behavior and neuronal activation in the medial prefrontal cortex and amygdala of male and female mice

PONE-D-22-19470R1

Dear Dr. Fadok,

We’re pleased to inform you that your manuscript has been judged scientifically suitable for publication and will be formally accepted for publication once it meets all outstanding technical requirements.

Kind regards,

Alexandra Kavushansky, PhD

Academic Editor

PLOS ONE

Additional Editor Comments (optional):

Reviewers' comments:

Reviewer's Responses to Questions

**Comments to the Author**

1. If the authors have adequately addressed your comments raised in a previous round of review and you feel that this manuscript is now acceptable for publication, you may indicate that here to bypass the “Comments to the Author” section, enter your conflict of interest statement in the “Confidential to Editor” section, and submit your "Accept" recommendation.

Reviewer #1: All comments have been addressed

2. Is the manuscript technically sound, and do the data support the conclusions?

Reviewer #1: (No Response)

3. Has the statistical analysis been performed appropriately and rigorously? 

Reviewer #1: (No Response)

4. Have the authors made all data underlying the findings in their manuscript fully available?

Reviewer #1: (No Response)

5. Is the manuscript presented in an intelligible fashion and written in standard English?

Reviewer #1: (No Response)

6. Review Comments to the Author

Reviewer #1: (No Response)

7. PLOS authors have the option to publish the peer review history of their article (what does this mean?). If published, this will include your full peer review and any attached files.

Reviewer #1: **Yes: **Mustapha Muzaimi

---

## [Editor Report · Acceptance letter]

30 Jan 2023

PONE-D-22-19470R1 

Effects of footshock stress on social behavior and neuronal activation in the medial prefrontal cortex and amygdala of male and female mice 

Dear Dr. Fadok:

I'm pleased to inform you that your manuscript has been deemed suitable for publication in PLOS ONE. Congratulations! Your manuscript is now with our production department. 

Kind regards, 

on behalf of

Dr. Alexandra Kavushansky 

Academic Editor

PLOS ONE